# Exploring the Gender and Age Demographics of Patients Treated by Emergency Medical Teams during Disasters

**DOI:** 10.3390/ijerph21060696

**Published:** 2024-05-28

**Authors:** Noriyuki Shiroma, Odgerel Chimed-Ochir, Yui Yumiya, Matchecane Cossa, Isse Ussene, Yoshiki Toyokuni, Kayako Chishima, Kouki Akahoshi, Seiji Mimura, Akinori Wakai, Hisayoshi Kondo, Yuichi Koido, Flavio Salio, Ryoma Kayano, Tatsuhiko Kubo

**Affiliations:** 1Department of Public Health and Health Policy, Graduate School of Biomedical and Health Sciences, Hiroshima University, Hiroshima 734-8553, Japan; d232539@hiroshima-u.ac.jp (N.S.); tkubo@hiroshima-u.ac.jp (T.K.); 2Ministry of Health of Mozambique, Maputo 1008, Mozambique; 3National Hospital Organization Headquarters DMAT Secretariat MHLW, Tokyo 190-8579, Japan; 4Ministry of Health, Labour and Welfare, Tokyo 100-8916, Japan; 5Emergency Medical Teams, World Health Organization, 1211 Geneva, Switzerland; 6World Health Organization Centre for Health Development (WHO Kobe Centre), Kobe 651-0073, Japan

**Keywords:** emergency medical team minimum data set, epidemiology, J-SPEED, disasters, health services accessibility, gender inequality

## Abstract

Background: Standardized health-data collection enables effective disaster responses and patient care. Emergency medical teams use the Japan Surveillance in Post-Extreme Emergencies and Disasters (J-SPEED) reporting template to collect patient data. EMTs submit data on treated patients to an EMT coordination cell. The World Health Organization’s (WHO) EMT minimum dataset (MDS) offers an international standard for disaster data collection. Goal: The goal of this study was to analyze age and gender distribution of medical consultations in EMT during disasters. Methods: Data collected from 2016 to 2020 using the J-SPEED/MDS tools during six disasters in Japan and Mozambique were analyzed. Linear regression with data smoothing via the moving average method was employed to identify trends in medical consultations based on age and gender. Results: 31,056 consultations were recorded: 13,958 in Japan and 17,098 in Mozambique. Women accounted for 56.3% and 55.7% of examinees in Japan and Mozambique, respectively. Children accounted for 6.8% of consultations in Japan and 28.1% in Mozambique. Elders accounted for 1.32 and 1.52 times more consultations than adults in Japan and Mozambique, respectively. Conclusions: Study findings highlight the importance of considering age-specific healthcare requirements in disaster planning. Real-time data collection tools such as J-SPEED and MDS, which generate both daily reports and raw data for in-depth analysis, facilitate the validation of equitable access to healthcare services, emphasize the specific needs of vulnerable groups, and enable the consideration of cultural preferences to improve healthcare provision by EMTs.

## 1. Introduction

Over 122 years, from 1900 to 2022, 5830 disasters triggered by natural hazards were recorded globally, with a quarter of the recorded events occurring in the last decade [1]. Small-scale disasters caused by climate change have also continued to seriously affect the health and economies of local communities in recent years [2].

Swift and coordinated responses from public emergency services are crucial to saving lives during disasters [3]. Real-time patient data, including demographics and health events, are essential for ensuring effective resource allocation to meet the increased medical demand that occurs during disasters [4,5]. Emergency medical teams (EMTs) consisting of doctors, nurses, paramedics, support workers, and logisticians are pivotal in delivering timely and high-quality health services during disasters to complement and support local health facilities. EMTs are categorized into four different types based on their mobility and level of care provided: Type 1 mobile, Type 1 fixed, Type 2 inpatient surgical emergency care, and Type 3 inpatient referral care [6]. The patient information gathered and analyzed by the deployed EMTs must serve the EMT Coordination Cell (EMTCC), which is typically administered by the Ministry of Health or an equivalent national authority. This information is essential for making necessary coordination and informed decisions and ensuring a timely and effective response [6]. However, until recently, EMTs have used different recording and reporting forms, hindering the compilation, analysis, and comparison of patient data across different disasters. Standardizing these forms is vital for improving the efficiency and effectiveness of disaster response efforts.

The Great East Japan Earthquake in 2011 also underscored the necessity for standardized EMT daily reporting procedures and led to establishment of the Joint Committee for Disaster Medical Recording and a proposal for Japan Surveillance in Post-Extreme Emergencies and Disasters (J-SPEED). J-SPEED was inspired by the Surveillance in Post-Extreme Emergencies and Disasters (SPEED) system developed in the Republic of the Philippines and was first activated during the 2016 Kumamoto Earthquake [7]. It currently serves as a comprehensive framework for EMTs in Japan to compile real-time health data during disasters. It has since been utilized in various disasters, including the Kumamoto and Hokkaido Eastern Iburi Earthquakes; Typhoons 15 and 19; and heavy rains in Western Japan, Northern Kyushu, and Kumamoto. Subsequently, in 2017, the World Health Organization (WHO) developed the Emergency Medical Team Minimum Dataset (MDS), an international version of J-SPEED [6]. The MDS was first activated during Cyclone Idai in Mozambique in 2019 [8]. Analysis of these data revealed that during disaster responses, EMTs tend to have more medical consultations with women and elders. However, comprehensive age and gender breakdowns across all the disaster events were not addressed. In addition, we were unable to find studies that comprehensively analyzed age and gender distribution of medical consultations by EMTs during disasters.

UNICEF reported that women, children, and youth are often recognized as being among the most vulnerable groups with regard to disasters triggered by natural hazards [9]. The WHO stated that EMT services should be accessible to all patients and members of the community based on their medical needs and not according to their ethnicity, religion, gender, age, political affiliation, or other factors. Access to health care in an emergency is a basic human right. EMTs need to ensure they are truly accessible to women, children, the elderly, those affected by disability, and those socially or financially disadvantaged [6]. It was also suggested that studying demographics ensures that emergency medical responses are equitable and inclusive. This helps in identifying disparities in access to healthcare services during disasters and guides efforts to provide fair and effective care to all individuals regardless of gender or age [10]. Understanding the gender and age distributions of those affected can help address specific needs and vulnerabilities and ensure that gender-inclusive disaster risk management and response practices are implemented [11]. Demographic data help in allocating resources effectively. By knowing the distribution of patients based on gender and age, EMTs can ensure they have the right equipment, medications, and personnel to address the diverse needs of the affected population. Analyzing demographic patterns from past disasters can also help EMTs to enhance their training, protocols, and response strategies to better serve the community in future disasters. 

Therefore, our analysis aimed to examine the gender and age distributions of medical consultations provided by EMTs during disasters by utilizing data collected through the J-SPEED and MDS tools during disasters that occurred from 2016 to 2020 in Japan and Mozambique. We also aimed to compare the age and gender distributions of medical consultations during Japanese and Mozambican disasters, as these distributions may vary significantly between countries and should be reflected in tailored emergency response strategies. 

## 2. Materials and Methods

### 2.1. Study Participants and Study Design

This study is a cross-sectional study. This study analyzed five disasters that occurred in Japan from 2016 to 2020: the Kumamoto [12] and Hokkaido Eastern Iburi Earthquakes [13], Typhoon 19 [14], and heavy rain in Western Japan [15] and Kumamoto [16]. Two disasters that occurred in Japan during the study period, the heavy rain in Northern Kyushu and Typhoon 15, were excluded from the analysis because of their shorter response periods and lower number of daily reports. The Kumamoto Earthquake, a 6.5 magnitude earthquake, struck on 14 April 2016, followed by a subsequent 7.3 magnitude earthquake on April 16. Up to 855 evacuation centers were opened, with the number of evacuees peaking at approximately 184,000. The Hokkaido Eastern Iburi Earthquake, a 6.7 magnitude earthquake, occurred on 6 September 2018, leading to landslides and a widespread blackout across Hokkaido. Typhoon 19 made landfall on 12 October 2019, affecting 230,000 people, with total precipitation exceeding 1000 mm in some areas from October 10th to 13th. The Western Japan heavy rain occurred on 8 July 2018, resulting in approximately 15,300 evacuees, 212 deaths, 342 injuries, and damage to 38,820 houses across Okayama, Hiroshima, and Ehime Prefectures. The Kumamoto heavy rain occurred on 4 July 2020, leading to around 11,000 evacuees, 67 deaths, 47 injuries, and damage to 7000 houses within Kumamoto Prefectures. 

Additionally, Cyclone Idai in Mozambique [17] was included in the analysis because available data for an international setting collected with the MDS tool was only available for the Mozambique cyclone. Cyclone Idai made landfall in the city of Beria in the province of Sofala on 15 March 2019. It affected 40,000 people. Cyclone Idai has been recognized as one of the deadliest cyclones on record for the Southern Hemisphere (See Table 1).

### 2.2. Data Collection

Japanese EMTs, including the Disaster Medical Assistance Team (DMAT), Japan Medical Association Team (JMAT), Japan Red Cross Medical Assistance Team (JRC), and Disaster Psychiatric Assistance Team (DPAT), provided medical consultations during each of the five disasters that occurred in Japan from 2016 to 2020. These EMTs were Type 1 mobile and Type 1 fixed.

Consultations reported by EMTs were reported daily in accordance with the J-SPEED form for disasters in Japan [18], while the WHO MDS tool was used to collect health-related data during Cyclone Idai in Mozambique [8]. The data recording form, data input method, and raw data generation process are identical for both the J-SPEED and MDS tools, with only a few differences in the items reported. The primary distinction is that J-SPEED is in Japanese, while MDS is in English. The J-SPEED and MDS tools generate daily reporting forms for coordination during disaster response and raw data in CSV (comma-separated value) format for detailed analysis. For this study, we utilized the raw data for our analysis. 

### 2.3. Data Analysis

In accordance with the study’s goals, information on the gender and age was extracted from the daily reports. The J-SPEED form included the age groups of <1, 1–14, 15–64, and >65 years, whereas the WHO MDS form included the age groups of <1, 1–4, 5–17, 18–64, and >65 years. We categorized the age groups of <1 and 1–14 (1–4 and 5–17 on the MDS form) as children, 15–64 (18–64 on the WHO MDS form) as adults, and anyone >65 years as elders. Subsequently, we conducted a descriptive analysis of the consultations according to age and gender. The ratio of consultation was calculated by comparing the consultation numbers between children and adults and elders and adults. The consultation rate was determined by dividing the total number of consultations by the population of the affected area [19,20]. The rate ratio, with reference to adults, was calculated by comparing the consultation rates among children and elders. As our primary goal is to understand overall trends of medical consultations rather than making predictions, we used simple regression analysis. We applied a 3-day moving average to remove potential random variations or noise that appear as coarseness in a plot of raw time series data through smoothing methods. The moving average can also remove a seasonal pattern to reveal underling trends. Regression analysis investigated the association between response days and consultation proportions in different genders and age groups. Considering the minimum response duration of 27 days across the five disasters (Table 1), the analysis covered only the initial 27 days. For Cyclone Idai in Mozambique, days 1–3 that encompassed zero consultations were excluded. Moreover, no consultations were reported for day 55; the analysis focused on days 4–54 of the response period. 

Microsoft Excel 2021 (Microsoft Corp., Redmond, Washington, DC, USA) and JMP v16 (SAS Institute Inc., Cary, NC, USA) were used for all analyses. 

## 3. Results

Table 1 shows general information on the disasters studied, including the number of evacuees, health events, and daily reports. The six disasters that occurred in Japan and Mozambique from 2016 to 2020 resulted in 1240 deaths and 5880 injuries, with 3114 daily reports collected using the J-SPEED or MDS forms. 31,056 consultations were recorded in total: 13,958 in Japan and 17,098 in Mozambique.

Table 2 outlines the demographic characteristics derived from the corresponding medical consultations. Women accounted for over half of the consultations in Japan (56.3%) and Mozambique (55.7%), even though boys sought EMT consultations more frequently than girls in both countries. Adults accounted for the largest proportion of consultations in both Japan (56.8%) and Mozambique (65.9%). Patients aged >65 years accounted for 36.4% of consultations in Japan, as compared to 6.0% in Mozambique. In Japan, children accounted for 6.8% of all consultations, as compared to 28.1% in Mozambique. The ratio of consultations between children and adults was 0.12 in Japan, and 0.43 in Mozambique. The consultation rate (per 100,000 people) in Japan was 65.39 for all ages, including 35.54 for children, 63.04 for adults, and 83.21 for elders. On the other hand, the consultation rate in Mozambique was 384.66 for all ages, including 232.34 for children, 502.04 for adults, and 760.78 for elders. The consultation rate for elders surpassed that for children and adults in both countries. 

Table 3 shows the age and gender distribution of earthquake disasters and heavy rain disasters. Adult women accounted for over half of the consultations in earthquakes (58.8%), as compared to 48.9% in heavy rain events. Patients aged over 65 years represented 31.8% of all consultations during earthquakes, contrasting with 44.3% in heavy rain situations. The consultation rate (per 100,000 people) was higher during earthquakes, with rates of 123.92 for all ages, 85.35 for children, 125.90 for adults, and 135.71 for elders. Conversely, during heavy rain disasters, the consultation rate was lower all ages (30.13 per 100,000 people), including rates of 10.03 for children, 26.23 for adults, and 47.52 for elders.

Figure 1 illustrates a comparative analysis of the observed trends regarding the consultation percentages across all patients, children, adults, and elders in Japan and Mozambique, with a focus on gender comparison. During the response period in Japan, a significant increase in the number of female consultations (β = 0.80, *p* < 0.0001) occurred, accompanied by a significant decrease in male consultations (β = −0.80, *p* < 0.0001). By contrast, no significant trends were observed in Mozambique. In Japan, there have been significant increases in the consultation proportions for women and decreases for men among adults and elders. Conversely, Mozambique experienced a significant decline in consultations for older women (β = −0.61, *p* < 0.0001) and a significant increase for older men (β = 0.61, *p* < 0.0001).

Figure 2 shows a comparative analysis of the observed trends regarding consultation percentages among men and women in Japan and Mozambique, with a focus on age comparison. In Japan, there was a significant increase in the number of adult consultations among men (β = 0.84, *p* < 0.0001) and women (β = 0.84, *p* < 0.0001). In contrast, a significant decrease was observed in the number of consultations for older men (β = −0.87, *p* < 0.0001) and women (β = −0.75, *p* < 0.001). Additionally, a decrease was noted in the number of consultations for girls (β = −0.57, *p* = 0.0026) in Japan.

In contrast, Mozambique showed a significant increase in the number of consultations for adults among men (β = 0.75, *p* < 0.0001) and women (β = 0.77, *p* < 0.0001). This was contrasted by a significant decrease in the number of consultations for boys (β = −0.80, *p* < 0.0001) and girls (β = −0.75, *p* < 0.0001) in Mozambique.

Similar patterns in consultation percentages were observed for both earthquake and heavy rain events (Appendix A). 

## 4. Discussion

We analyzed 3114 daily medical reports collected during the six documented disasters in Japan and Mozambique from 2016 to 2020, focusing on the gender and age distribution of the 31,056 health consultations provided by EMTs. In both countries, more than half of the consultations involved women. Among children, boys in Japan and Mozambique had a higher proportion of health consultations. Elders in Japan visit EMTs more frequently than children, whereas in Mozambique, EMTs tended to receive more children than elders. After adjusting for the age composition of the two countries, we found that the consultation rate for elders surpassed that for children and adults in both Japan and Mozambique.

Our results align with those of previous studies. For example, a report on the Great East Japan Earthquake in 2011 revealed that 54% of the people receiving medical care at evacuation centers were women, excluding those of unknown gender [21]. Similarly, Cuesta et al. reported a higher frequency of medical treatment for women than men (55.9% vs. 44.1%, respectively) at a field hospital during Typhoon Haiyan in the Philippines in 2013 [22]. However, these studies possessed a number of limitations, which included utilizing only a fraction of the collected data, employing different data collection methods, and gathering information from different locations, such as field hospitals and evacuation shelters. These disparities pose challenges in terms of drawing comparisons among these studies. Thus, our study corroborates the results of these earlier studies by systematically analyzing data collected daily by EMTs. The results obtained suggest a higher likelihood of women seeking medical care from EMTs during disasters. However, there is no concrete evidence to explain this trend. It is possible that because the majority of men are still working during such disasters; this means that women are more likely to visit medical facilities. However, other studies have proposed different interpretations, with some suggesting that women engage in better health-seeking behaviors because of their higher health literacy [23]. The Sendai Framework for Disaster Risk Reduction emphasizes how women experience disasters differently, highlighting their increased vulnerability to specific disaster situations [2]. Miki et al. reported higher mortality rates among women and children during disasters [24], whereas others reported higher vulnerability among men [25]. Therefore, these results should be interpreted with caution in the context of disasters. Although both genders might have required equal medical care, factors such as the prioritization of women over men with regard to the provision of medical services or men opting to avoid long queues at medical facilities may have influenced the observed pattern.

In our comparative analysis between Japan and Mozambique, older patients (>65 years) had considerably higher consultation rates than children in both countries. This finding demonstrates that older adults are more likely to visit EMTs. However, a distinct contrast emerged between the two countries. In Japan, older patients accounted for a substantially higher proportion of EMT visits than children, whereas in Mozambique, children were more likely to be treated by EMTs than older patients. This finding suggests that EMTs should consider the demographic characteristics of a given country during the preparedness and mobilization phases because children and elders require different specialties, skills, medical supplies, and equipment. Understanding and adapting to these demographic variations can ensure that EMTs offer more efficient and effective disaster responses tailored to each country’s specific healthcare needs.

This study’s findings have several implications. The key advantage of this research is the data gathering systems, J-SPEED and MDS. Real-time data reporting allows the EMTCC to conduct data-driven coordination throughout the response. As a result, the utilization of the J-SPEED and MDS enables the identification of areas and timeframes with lower rates of female consultations, supporting the EMTCC in coordinating targeted interventions. These data can be used to validate and showcase equitable access to healthcare services. Although the results of the current study revealed that women were more likely to receive treatment from EMTs, further research is warranted to understand and address potential gender disparities in healthcare-seeking behaviors. Second, by leveraging the demographic information of affected populations, EMTs can better estimate and organize medical supplies during deployment, emphasizing the specific needs of vulnerable groups such as children and older patients. Finally, increased awareness of patient demographics is crucial for EMTs, especially for providing medical services tailored to female patients. For instance, in Islamic countries, cultural and religious preferences regarding healthcare providers indicate a preference for female doctors when providing care to Muslim women. Islamic bioethics underscores a hierarchical preference. Muslim female healthcare providers are favored as the primary choice for Muslim female patients, followed by non-Muslim female healthcare providers [26]. Additionally, research has shown that the gender of EMT members can be an important factor associated with patients refusing medical aid, with a higher percentage of refusal occurring when the gender of the EMT members does not match that of the patient [27]. Therefore, EMTs should consider and prepare for such cultural nuances when engaging in healthcare provision.

The current study could have several policy implications. Disaster management is a multifaceted endeavor requiring coordination among various stakeholders, including government agencies, non-governmental organizations, healthcare providers, and local communities. Understanding the demographic patterns of health consultations during disasters can inform resource allocation, training programs, and public health initiatives to enhance the effectiveness and equity of emergency medical services. Moreover, one significant implication of the study is the need for gender-sensitive disaster response strategies. The disproportionate utilization of emergency medical services by women underscores the importance of addressing gender-specific vulnerabilities and healthcare needs during disasters. This requires integrating gender-sensitive approaches into disaster risk reduction policies and emergency response plans. Furthermore, the study highlights the importance of considering age-specific healthcare requirements in disaster planning. The higher consultation rates among older adults emphasize the need for specialized medical care and support for this demographic group during emergencies. This includes ensuring access to essential medications, medical equipment, and geriatric care services in disaster response efforts. Similarly, addressing the unique needs of children, such as pediatric medical supplies and psychological support, is critical for promoting their health and well-being during and after disasters. Moreover, the study underscores the significance of data-driven decision-making in disaster management. By leveraging standardized data collection tools such as the J-SPEED and MDS, emergency response organizations can monitor healthcare utilization patterns, identify vulnerable populations, and allocate resources more effectively.

This study has several strengths and limitations. To the best of our knowledge, this is the first study to investigate these five disasters that occurred in Japan using the J-SPEED uniform data collection form. This standardized form enabled the effective compilation and comparison of data across all the disasters studied. Another strength of this study is the comparison of disasters in both Japan and international setting. Moreover, we opted for a pooled data approach merging data of five disasters in Japan to ensure more robust findings. By merging data from various disasters, our study endeavors to offer a comprehensive and generalizable understanding of age and gender distribution in medical consultations during emergencies. This approach facilitated: (i) identification of common trends of disasters; (ii) enhancement of the robustness and reliability of findings through increased statistical power; (iii) mitigation of outliers and biases in single disaster, ensuring balanced and accurate conclusions.

The study also possesses a number of limitations. First, the J-SPEED only reports information from EMTs, excluding data from other health facilities. Thus, the data obtained may not reflect a complete picture of the entire situation. Secondly, there is the possibility of under-reporting, as some patients may not have been documented during the disasters. However, no selection bias was found in the patient recordings, as there was no prioritization of female patients. Third, owing to the lack of information on evacuees, population data from areas beyond those affected were used to calculate the consultation rates. This may limit the accuracy of the comparison of the consultation rates between both countries. Fourth, the study was not adjusted to account for additional demographic and economic factors that may introduce bias [28] concerning required access to care by EMTs. Given these findings and limitations, further research is needed to investigate the economic and social factors that influence health-seeking behaviors among all age groups during the initial phases of disasters. Further investigation is also required to explore the specific medical care needs of men and women during disasters. Finally, findings were not attainable for each disaster alone due to a limited number of data. Analyzing the causes necessitates evaluating each disaster in conjunction with associated diseases, and further data accumulation will be imperative for future studies.

## 5. Conclusions

In both Japan and Mozambique, a higher proportion of female patients sought medical assistance from EMTs compared to male patients. Moreover, adult patients constituted the majority of EMT visits in both countries, followed by older patients in Japan and children in Mozambique. These findings highlight the importance of considering age-specific healthcare requirements in disaster planning. In addition, real-time data collection tools such as J-SPEED and MDS generate both daily reports and raw data for in-depth analysis, facilitate the validation of equitable access to healthcare services, emphasize the specific needs of vulnerable groups, and enable the consideration of cultural preferences to improve healthcare provision by EMTs. Future studies should focus on assessing the actual medical needs of diverse demographic groups by considering not only demographic variables but also socioeconomic variables and health-seeking behavior patterns.

## Figures and Tables

**Figure 1 ijerph-21-00696-f001:**
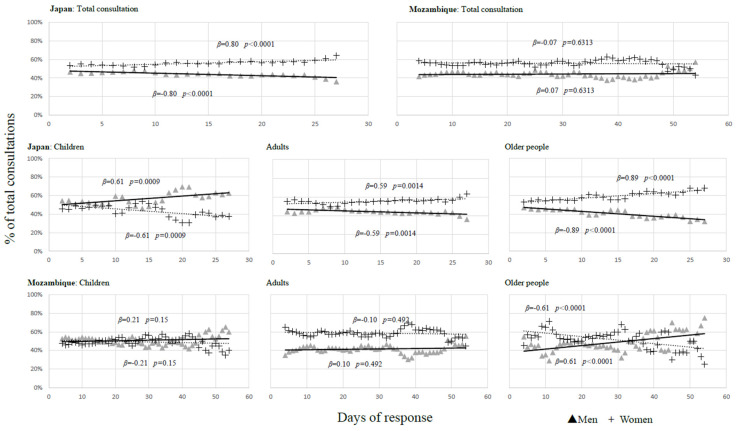
Trend of medical consultation by gender comparison.

**Figure 2 ijerph-21-00696-f002:**
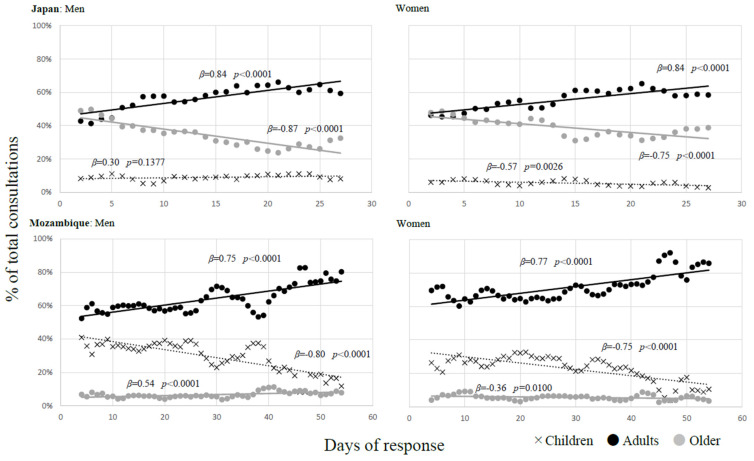
Trend of medical consultation by age comparison.

**Table 1 ijerph-21-00696-t001:** General information of the studied disasters.

	Kumamoto Earthquake	Western Japan Heavy Rain	Hokkaido Eastern Iburi Earthquake	Typhoon 19	Kumamoto Heavy Rain	Mozambique Cyclone IDAI	Total
Affected prefecture	Kumamoto	Hiroshima, Okayama, Ehime	Hokkaido	Nagano, Miyagi, Fukushima, Ibaraki	Kumamoto	Sofala, Manica	
Evacuees ^1^	184,000	15,300	17,000	237,000	11,000	400,000	864,300
Number of deaths ^1^	228	212	42	91	67	600	1240
Number of injured ^1^	2753	342	762	376	47	1600	5880
Houses damaged ^1^	200,000	38,820	14,600	96,500	7000	240,000	596,920
Data collection period ^2^	16 April–2 June 2016	8 July–11 September 2018	6 September–7 October 2018	13 October–21 November 2019	5 July–31 July 2020	27 March–12 July 2019	
Days of the response ^2^	48 days	65 days	32 days	40 days	27 days	110 days	
Number of daily reports ^2^	1830	402	191	201	208	282	3114
Total number of patient consultations ^2^	8102	3617	739	684	816	17,098	31,056

^1^ Information source is the Cabinet Office, Government of Japan, White Paper on Disaster Management. ^2^ Summarized from the J-SPEED form.

**Table 2 ijerph-21-00696-t002:** Age and gender distribution of medical consultations provided during the studied disasters: comparison of all disasters in Japan and Cyclone Idai in Mozambique.

Age and Gender	Japan				Mozambique			
	Number of Consultations (%)	Ratio of Number of Consultations ^2^	Consultation Rate ^3^	Rate Ratio ^4^	Number of Consultations (%)	Ratio of Number of Consultations ^2^	Consultation Rate ^3^	Rate Ratio ^4^
All ages	13,958 (100)		65.39		17,098 (100)		384.66	
Men	6101 (43.7)		59.53		7570 (44.3)		352.57	
Women	7857 (56.3)		70.80		9528 (55.7)		414.65	
Children ^1^	944 (6.8)	0.12	35.54	0.56	4798 (28.1)	0.43	232.34	0.46
Men	501 (53.1)		36.84		2404 (50.1)		233.62	
Women	443 (46.9)		34.19		2394 (49.9)		231.07	
Adults ^1^	7937 (56.8)	Ref	63.04	Ref	11,265 (65.9)	Ref	502.04	Ref
Men	3529 (44.5)		56.07		4694 (41.7)		443.54	
Women	4408 (55.5)		70.02		6571 (58.3)		554.26	
Elders ^1^	5077(36.4)	0.64	83.21	1.32	1035 (6.0)	0.09	760.78	1.52
Men	2071 (40.8)		79.83		472 (45.6)		789.28	
Women	3006 (59.2)		85.71		563 (54.4)		738.43	

^1^ Children refers to <1 year old, 1–4, 5–17 in MDS form and 0–14 years in J-SPEED form. Adult refers to age of 18–64 in MDS form and 15–64 in J-SPEED form. Elders refers to 65 years old and over in both MDS and J-SPEED forms. ^2^ Ratio of number of consultations among children and adults as well as elders and adults. The reference group is adults. ^3^ Number of consultations per 100,000 population of all affected prefectures. ^4^ Ratio of consultation rate of children and adults as well as elders and adults. Reference group is adults.

**Table 3 ijerph-21-00696-t003:** Age and gender distribution of medical consultations provided during the studied disasters: A comparison between earthquake and heavy rain events.

Age and Gender	Earthquake				Heavy Rain			
	Number of Consultations (%)	Ratio of Number of Consultations ^2^	Consultation Rate ^3^	Rate Ratio ^4^	Number of Consultations (%)	Ratio of Number of Consultations ^2^	Consultation Rate ^3^	Rate Ratio ^4^
All ages	8841 (100)		123.92		5117 (100)		30.13	
Men	3571 (40.4)		106.29		2530 (49.4)		30.75	
Women	5270 (59.6)		139.60		2587 (50.6)		29.56	
Children ^1^	725 (8.2)	0.14	85.35	0.68	219 (4.3)	0.08	10.03	0.38
Men	378 (52.1)		87.10		123 (56.2)		10.99	
Women	347 (47.9)		83.51		96 (43.8)		9.02	
Adults ^1^	5307 (60.0)	Ref	125.90	Ref	2630 (51.4)	Ref	26.23	Ref
Men	2186 (41.2)		106.00		1343 (51.1)		26.58	
Women	3121 (58.8)		144.96		1287 (48.9)		25.88	
Elders ^1^	2809 (31.8)	0.53	135.71	1.08	2268 (44.3)	0.86	47.52	1.81
Men	1007 (35.8)		116.63		1064 (46.9)		51.76	
Women	1802 (64.2)		149.37		1204 (53.1)		44.32	

^1^ Children refers to <1 year old, 1–4, 5–17 in MDS form and 0–14 years in J-SPEED form. Adult refers to age of 18–64 in MDS form and 15–64 in J-SPEED form. Elders refers to 65 years old and over in both MDS and J-SPEED form. ^2^ Ratio of number of consultations among children and adults as well as elders and adults. The reference group is adults. ^3^ Number of consultations per 100,000 population of all affected prefectures. ^4^ Ratio of consultation rate of children and adults as well as elders and adults. Reference group is adults.

## Data Availability

Restrictions apply to the availability of these data. The data used in this study were obtained from the J-SPEED Research Group for research purposes and were made available with the permission of the group.

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
