# Peer review of "Exploring the Gender and Age Demographics of Patients Treated by Emergency Medical Teams during Disasters"

_ijerph, 2024, doi:10.3390/ijerph21060696_

Round 1
Reviewer 1 Report
Comments and Suggestions for Authors
The article “Exploring the Sex and Age Demographics of Patients Treated by Emergency Medical Teams during Disasters” addresses a relevant thematic area. However, it presents several limitations that need to be addressed.
Major points.
-From the initial reading, a significant concern arises regarding the inclusion of the Mozambique cyclone among the studied disasters. What scientific rationale was adopted for this inclusion? What is the relevance of the comparison between Japan and Mozambique? Why were these countries chosen and not others as well?
-The presented results are overly simplistic, focusing solely on the distribution of records by sex and age. Are there no other variables to be studied? If so, why were they not included? The inclusion of more information would enrich the study.
-Another significant point concerns the presentation of results. While five different events in Japan are analyzed, the results are presented in a grouped manner. Separate presentation would offer much more useful and relevant information. I consider this correction essential for justifying the study publication.
-Furthermore, both the conclusion presented in the abstract and the article itself are vague and lack specificity regarding the practical implications for decision-making arising from the results.
-Considering the aim of comparing Japan and Mozambique (which was not adequately justified), the impact of the different methods of data collection in the two countries should be discussed.
-There is no presentation or discussion of ethical issues.
Other points
-The abstract suffers from an excessive use of acronyms, hindering comprehension. Moreover, it lacks a clear criterion justifying the research and the analytical procedures conducted.
-The introduction provides better contextualization and justification for the research; however, it fails to clearly identify existing studies similar and what the study adds to the literature. Moreover, there is no presentation or discussion of ethical issues.
-In the methods section, a more detailed presentation of the study areas and the disasters studied is necessary. The inclusion of one or more maps would be beneficial.
-The results section, Table 2 and the figures are not self-explanatory. Clear explanations are needed.
Reviewer 2 Report
Comments and Suggestions for Authors
This is an interesting manuscript that examines age and sex distributions of health consultations during disaster responses. This manuscript analyzes the time effect using linear regression models. This is a significant limitation as there are specific methods to analyze time series. The authors conclude that J-SPEED and WHO MDS can support effective monitoring in disaster responses, but this work does not aim to analyze the validity of these instruments. The Conclusion has to be in line with the objective of the paper.
Reviewer 3 Report
Comments and Suggestions for Authors
Interesting manuscript, but that can be improved:
In the introduction more data about EMT classification should be described. Also, what type of EMT do the authors refer to? In different types, this should be presented in the results; different EMT types may have different patient profiles
- objetives are not clear defined
Round 2
Reviewer 1 Report
Comments and Suggestions for Authors
The authors responded appropriately to my comments and justified the choices they made.
Reviewer 2 Report
Comments and Suggestions for Authors
The manuscript has been improved in the updated version